# MaskID: An effective deep-learning-based algorithm for dense rebar counting

**Wenrui Li[2], Jian Cheng[2], Bo Chen[2], Yu Xue[2], Yi Wang[3], Yan Fu[1,4], Junlin Zhou[1,4], Duanbing Chen** [1,4]*

1 School of Computer Science & Engineering, University of Electronic Science and Technology of China, Chengdu, P.R. China, 2 The First Construction Engineering Limited Company of China Construction Third Engineering Bureau, Wuhan, P.R. China, 3 Sichuan University of Science & Engineering, Yibin, P.R. China, 4 Chengdu Union Big Data Tech. Inc., Chengdu, P.R. China

* dbchen@uestc.edu.cn

## Abstract

As a dense instance segmentation problem, rebar counting in a complex environment such as rebar yard and rebar transpotation has received significant attention in both academic and industrial contexts. Traditional counting approaches, such as manual counting and machine vision-based algorithms, are often inefficient or inaccurate since rebars with varied sizes and shapes are stacked overlapping, rebar image is not clear for complex light condition such as dawn, night and strong light, and other environmental noises exist in rebar image; thus, they no longer fulfil the requirements of modern automation. This paper proposes MaskID, an innovative counting method based on deep learning and heuristic strategies. First, an improved version of the Mask region-based convolutional neural network (Mask R-CNN) was designed to obtain the segmentation results through splitting and rescaling so as to capture more detail in a large-scale rebar image. Then, a series of intelligent denoising strategies corresponding to aspect ratio of recognized box, overlapping recognized objects, object distribution and environmental noise, were applied to improve the segmentation results. The performance of the proposed method was evaluated on open-competition and test-platform datasets. The $F_1$-score was found to be over 0.99 on all datasets. The experimental results demonstrate that the proposed method is effective for dense rebar counting and significantly outperforms existing state-of-the-art methods.

## Introduction

### Motivation

Up to now, several in-depth studies of the rebar counting problem have obtained good results. However, certain factors make accurate counting a challenging task. These include the following:

1. Counting problems with densely stacked rebars: overlapping and unclear borders make accurate recognition very difficult.

**Funding:** This work was partially supported by the National Natural Science Foundation of China

(Grant No. 61673085), by the Science Strength Promotion Program of UESTC under Grant No. Y03111023901014006. There was no additional external funding received for this study.

**Competing interests:** The authors have declared that no competing interests exist.

2. Environmental constraints: several challenges are faced in obtaining clear and usable images for accurate counting under different lighting conditions, such as dawn, day, and night.

3. Various sizes of rebars to be detected: the rebars have a wide range of diameters; moreover, in practice, the shape of the rebar may be irregular.

Therefore, to enable accurate counting in a complex environment, this paper proposes a novel deep-learning-based rebar recognition and heuristic strategy-based counting approach. Two main principles of the approach are as follows. Firstly, splitting and rescaling are introduced in the original Mask R-CNN [1] to capture more details of large-scale images and an improved version of Mask R-CNN is designed. Secondly, to achieve highly accurate recognition results, multiple intelligent denoising strategies corresponding to aspect ratio of recognized box, overlapping recognized objects, object distribution and environmental noise, are designed and applied to remove the noise in the results obtained by improved Mask R-CNN.

## Background and related works

The rebar counting problem essentially calls for the segmentation of densely stacked objects. The earliest counting machine with an automatic sorting mechanism originated in Japan; it was a dual-transmission mechanical system realized by phototubes, which could handle situations such as bending, overlapping, and shielding of bar-like materials. Offoiach [2] attempted to improve automatic counting by using both impulsers and phototubes, but this did not achieve the expected outcome; the rebars were only recognized sequentially. Unexpected situations, such as bending, could result in undercounting or miscounting. Dynamic Ventures Inc. developed a commercial APP called CountThings (https://www.countthingsfromphotos.com/) to count rebars, logs, pipes, etc. For a typical image, rebars can be counted with low miscounting errors. However, for certain scene, template should be developed while counting the objects in an image.

In recent years, several deep-learning-based approaches have been developed for object detection. In 2012, AlexNet [3] was applied to large-scale object classification for the first time, and it won the ILSVRC-2012 image classification competition. AlexNet achieved a winning top-5 test error rate of 15.3%, compared to 26.2% achieved by the second-best entry. Thus far, most object detection methods have been anchor-based [4]. They can be broadly classified into two categories: region proposal–based object detection (two-stage structure) and regression-based object detection (one-stage structure). The representational networks of a two-stage structure include the region-based convolutional neural network (R-CNN) [5] and its improved versions such as Fast R-CNN [6], Faster R-CNN [4, 7], Cascade R-CNN [8] and Mask R-CNN [1]. Retina Net [9] and YOLO [10] are representation of regression-based networks. These algorithms exhibit excellent performance in general scenarios. However, they frequently fail in special cases, such as those of overlapping or dense objects. Very recently, densely packed object detection has received significant attention. Precise detection in densely packed scenes (PDDPS) [11] is the first network that aimed for dense object detection. It introduced Gaussian mixture models to handle overlapping objects and was shown to be highly effective. Ding et al. [12] introduced the RoI transformer to handle the misalignment of densely stacked objects in aerial images. Moreover, Zhu et al. [13] created an object detector, known as *ScratchDet*, which took advantage of the Root-ResNet of the original image to enhance the recognition capability of small objects in combination with ResNet and VGGNet. In 2020, an increasing number of studies have focused on densely stacked object detection. Yang et al. [14] proposed dense repoints, a creative object representation realized by a dense

point set at multiple levels, which strengthens the performance of contour-based counterparts. Chu et al. [15] presented the concept of allowing one proposal to be responsible for a set of correlated instances, which gains 4.9% average precision (AP) elevation using Earth Mover's Distance (EMD) and non-maximum suppression (NMS). Furthermore, Pan et al. [16] presented a dynamic refinement network which can dynamically optimize classification tasks using two novel components: a feature selection module(FSM) and a dynamic refinement head (DRH). FSM enables neurons to adjust receptive fields in conformity with the shapes and orientations of the targets. On the other hand, DRH allows the model to dynamically improve the prediction. Qiu et al. [17] designed BorderDet, an innovative detection architecture which utilizes border information for more persuasive classification and precise localization. It obtains a 50.3% AP and outperforms the majority of recent approaches with the ResNeXt-101-DCN backbone. Xie [18] proposed a semantic segmentation and keypoint detection algorithm based on weak supervision to count rebars. It shows superior performance compared to that of Faster R-CNN and cascaded R-CNN. Lee et al. [19] presented a robust algorithm for detecting small and dense objects in images from autonomous aerial vehicles based on cascaded R-CNN and recursive feature pyramid. It won the VisDrone-DET 2020 challenge with 34.57% mAP, compared to 34.54% achieved by the second-best entry. Zhao et al. [20] presented a knowledge-aided CNN for small organ segmentation with limited training data using two cascade steps, i.e., localization and segmentation. The results are rather good on classical datasets such as ISBI 2015 VISCERAL.

## Key contributions

The performance of the proposed method was evaluated on datasets of open competition and a test platform built for this purpose. The results demonstrate that it significantly outperforms state-of-the-art methods.

The key contributions of the paper are as follows:

- A novel algorithm MaskID is proposed to count the rebars in complex environment.

- An improved version of Mask R-CNN is presented through splitting and rescaling so as to capture more details in a large-scale rebar image.

- Intelligent denoising strategies are designed to remove the noise in the results obtained by improved Mask R-CNN.

- Experiments on open-competition and test-platform datasets show that MaskID can obtain more accurate counting results than other state-of-the-art methods.

## Organization of the paper

The remainder of this paper is organized as follows. Section 2 describes the rebar counting method based on the improved version of Mask R-CNN and heuristic denoising strategies. Section 3 presents the experimental results and discussions on open-competition and test-platform datasets. Finally, Section 4 presents the conclusions and future works.

## Materials and methods

### Mask R-CNN

For a rebar image, if a series of region proposals can be used to accurately frame each rebar in image, the number of rebars can be achieved. Because the region proposal extracted from

model may have a large deviation from the real bounding box of rebar, e.g., IoU < 0.5, it is necessary to fine tune region proposal to make the fine tuned bounding box closer to the real bounding box than original region proposal. The bounding box regression is used to fine tune the region proposal in deep learning method.

As previously illustrated, deep-learning models are backbones in instance segmentation. Some excellent methods have been presented to segment objects in various scene. Wu et al. [8] proposed a multiple attention encoded cascade R-CNN to detect scene-text in complex natural scene which progressively refines accurate boundaries of text instances. In their method, two core stages are included, i.e., feature generation and cascade detection. Wan and Goudos [7] proposed a faster R-CNN to detect multi-class fruit for facilitating high level smart farm. Three key strategies are included in their method, i.e., fruit image library creation, data argumentation and improved faster R-CNN model generation. Among all the models, Mask R-CNN [1] was selected as the backbone model in our approach. This is because Mask R-CNN outper- forms the majority of other models in instance segmentation tasks, which is highly significant for small, intensively-stacked object detection. As an extension framework of Faster R-CNN [4], Mask R-CNN completes the instance segmentation task by adding a mask prediction branch in parallel with the existing branch for bounding box regression. Because RoIPool in Faster R-CNN is not a pixel-to-pixel alignment, the quantization operation will significantly increase the mask error rate. To address this issue, RoIAlign was introduced in Mask R-CNN to replace the RoIPool operation, enhancing the mask accuracy by using a bilinear interpola- tion strategy [16]. The structure of the Mask R-CNN used in this study is shown in Fig 1. In this structure, convolutional neural network and region proposal network are utilized to extract the feature map and to obtain the region proposals of each object(Fig 1(2)) from origi- nal image firstly, based on this, fixed size feature map (Fig 1(3)) is obtained through RoIAlign layer, finally, bounding box of object is obtained by bounding box regression (Fig 1(4)). To better understanding the Mask R-CNN, several details are elaborated. For model architecture, Mask R-CNN with ResNeXt101 [21] backbone with four stages is adopted and the backbone is refined by replacing standard convolutional layers with deformable convolutional layers in last three stages. Besides, in the first stage, multiple candidate bounding boxes are proposed to clas- sify foreground and background and to regress bounding-box coordinate offsets. In the second

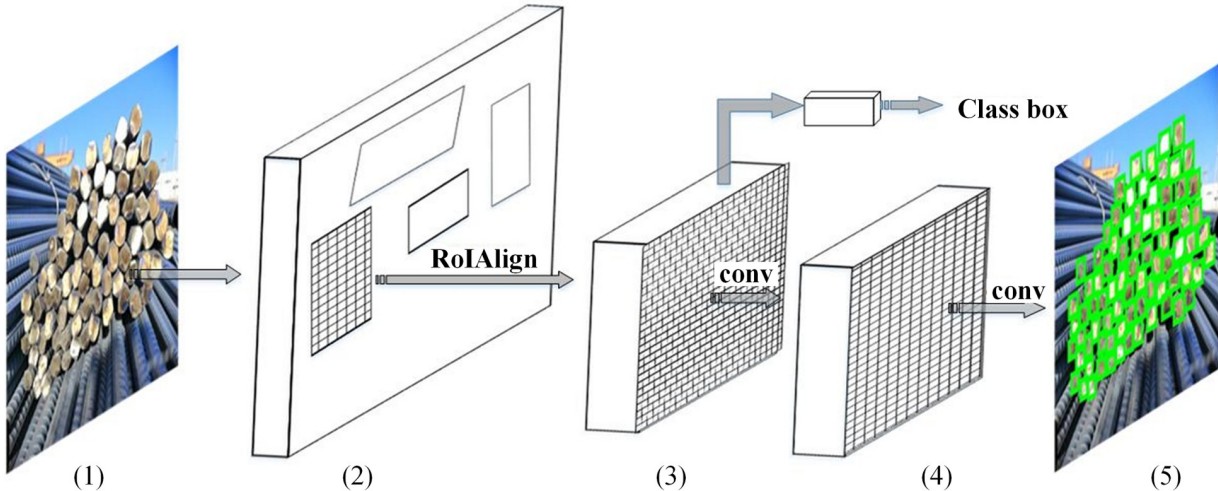

**Fig 1. (Color online)Mask R-CNN framework.** (1) original rebar image, (2) feature map and region proposals, (3) fixed size feature map, (4) bounding box regression, and (5) final bounding box of rebar.

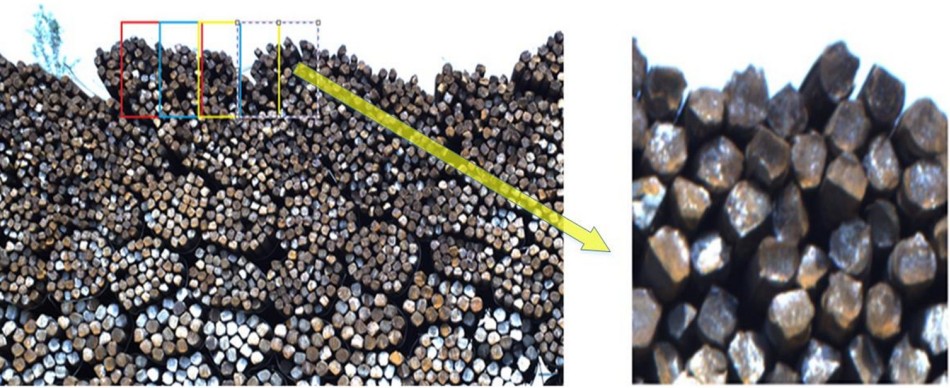

**Fig 2. (Color online)Schema for splitting and rescaling.** The left image is the original rebar image, and the right image is a slice from the original image that rescaled to 1024 × 1024 pixels.

stage, features are extracted using RoIAlign from each candidate box and performs classification and bounding-box regression in parallel. The Mask R-CNN is trained by using SGD optimizer with 0.02 learning rate.

In the case of dense rebar counting, owing to the large difference in the sizes of rebars in an image, the original Mask R-CNN is not very effective for detection, because some small-sized rebars are not very clear. Thus, to effectively detect rebars of various sizes, it is necessary to split a large image into several smaller images before applying Mask R-CNN. So, each smaller image is an input in the Mask R-CNN (Fig 1(1)) in this study, the counting results of large image can be obtained by fusing the counting results of each smaller image. To retain the original information of the image, a large image is split into several overlapping small images, which guarantees that the model can inspect the intact image of rebar sections. In this study, the original rebar image is compartmentalized into 384 × 384-pixel small images with 160 pixels overlapping to retain the complete information of the object. After being split, each small image is rescaled to 1024 × 1024 pixels to obtain more feature information. The schema for splitting and rescaling are shown in Fig 2.

## Intelligent denoising strategies

Because overlapping regions exist in split images, some rebars are counted repeatedly in multiple split images and some background noise is identified as rebar, the recognition results should be processed further to eliminate noise.

**Noise detection based on aspect ratio of a box.** Because the cross-section of a rebar is generally approximated by a circle, the length and width of the recognized box will not differ significantly. If the aspect ratio of the recognized box is below a certain threshold, it indicates that the recognized box is abnormal and needs to be removed. After removing boxes with abnormal aspect ratios, inscribed circles of boxes were obtained as candidate objects.

**Noise detection based on two overlapping objects.** For a dense rebar stacking scenario, the sizes of two adjacent rebars do not differ significantly. For this reason, any two overlapping recognized objects need to be further processed based on the specific situation in order to remove noise.

### (1) An object is entirely contained in another object.

For this specific situation, the smaller object is most likely to be noise and needs to be removed. As shown in Fig 3(a), the smaller light gray circle will be removed.

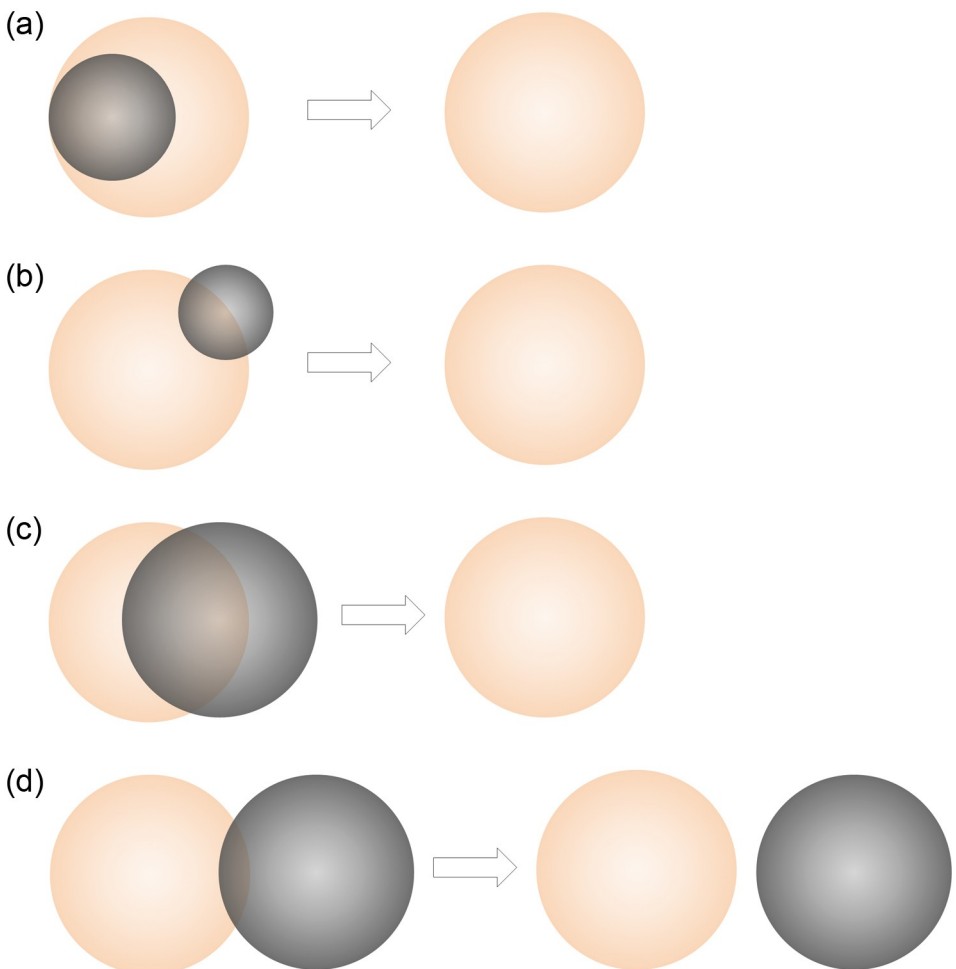

**Fig 3. (Color online)Mechanism of eliminating overlapping region.** In (a), (b), and (c), the light gray circles are noise, and in (d), both circles are real rebars and will be retained. (a) An object is entirely contained in another object. (b) One object is significantly smaller than another. (c) Two objects of similar size with large embedding. (d) Two objects of similar size with small embedding.

**(2) One object is significantly smaller than another**.

   If the size of the larger object is basically the same as that of other rebars around it, the smaller object is most likely noise and needs to be removed. As shown in Fig 3(b), the smaller light gray circle will be removed.

**(3) The objects are similar in size**.

   If the distance between their centers is below a certain threshold $\delta$ ($\delta < 0.5(r_1 + r_2)$, where $r_1$ and $r_2$ are the radii of two circles, respectively), the object with lower reliability (each recognized object predicted by Mask R-CNN has a score reflecting its prediction reliability) will be removed. As shown in Fig 3(c), the light gray circle with lower reliability will be removed. If the distance between their centers is larger than $\delta$, both candidate objects will be retained. As shown in Fig 3(d), both the light orange and light gray circles will be retained.

**Noise detection based on object distribution.**　For dense rebar recognition, the size of a recognized object is approximately the same as that of the surrounding objects. If the area of the currently recognized object $s_i$ is significantly smaller or larger than the average area of the surrounding objects $\langle s \rangle$ ($\frac{s_i}{\langle s \rangle} < 0.2$ or $\frac{\langle s \rangle}{s_i} < 0.2$ in this study), the current recognized object is considered to be noise and needs to be removed. In addition, if the area of the current object $i$ is significantly smaller than that of the surrounding objects, object $i$ will be removed. Essentially, for any adjacent object $j$ where the distance to object $i$ is less than $k * r_i$, where $k$ is a tunable parameter between 5 to 10 (the noise cannot be removed for too much $k$ and the real rebar might be removed for too small $k$, $k = 8$ in this study) and $r_i$ is the radius of $i$, if the area $s_j$ of object $j$ is greater than $m * s_i$, where $m$ is a tunable parameter between 3 to 7 (the noise cannot be removed for too much $m$ and the real rebar might be removed for too small $m$, $m = 5$ in this study) and $s_i$ is the area of object $i$, object $i$ is considered to be noise and needs to be removed.

**Environmental noise detection.**　For an image of dense rebars, some regions that are distant from actual rebars may be mistakenly identified as rebars because of the influence of environmental noise. The following rule is used to check for this case: in a certain region, if the total area of objects $s_{sum}$ is significantly smaller than the area of the envelope rectangle $s_{rec}$ of these objects ($s_{sum} < 0.05 s_{rec}$ in this study), all objects in this region are considered to be noise and are removed.

## Model training and testing

In the training process, each original training image is split into small images for labeling. The total number of small images is about 4000, and the number of marked rebars is more than 50000. The labeled samples are input to improved Mask R-CNN for training model with SGD optimizer and 0.02 learning rate to obtain the rebar recognition model. In rebar recongnition process, trained improved Mask R-CNN model is used to identify the rebar firstly, and heuristic denoising strategies are applied to remove the noise to get the final recognition results.

## Results and discussion

### Datasets

The performance of the proposed method was evaluated using two datasets. One is provided by the DataFountain competition, which contains 450 images where 250 images are training set and 200 images are testing set. All images were captured using smartphones and can be obtained from https://www.datafountain.cn/competitions/332/datasets. This dataset possesses some typical features that are described as follows. (1) Rebars are densely stacked, which increases the difficulty of segmentation. (2) The rebar sizes vary significantly. (3) There are notable differences in the angles from which the photographs were captured.

The other dataset is from a physical test platform established in this study, which contains approximately 8000 rebars of various sizes. The physical test platform is constructed in the building site, which is similar to a truck loaded with densely stacked rebars and can be used to simulate the actual complex rebar counting scenario.

In this paper, 250 training images from DataFountain competition are used to train MaskID. 200 testing images from DataFountain competition and one image with $6405 \times 3597$ pixels obtained from physical test platform are used to evaluate the performance of MaskID.

### Evaluation metrics

Because the rebar recognition problem can be considered as an object classification task in the field of machine learning, Precision, Recall and $F_1$-score are generally used for evaluation.

**Table 1. Confusion matrix.** TP denotes the number of prediction objects that are real rebars, FP denotes the number of prediction objects that are not real rebars, and FN denotes the number of real rebars that are not recognized.

|  | Actual Positive | Actual Negative |
|---|---|---|
| Predict Positive | TP | FP |
| Predict Negative | FN | TN |

First, the confusion matrix was created as shown in Table 1. In this table, precision *Pr* and recall rate *Re* can be calculated using the confusion matrix, as shown in Eq 1.

$$\left.\begin{aligned} Pr &= \frac{TP}{TP + FP} \\ Re &= \frac{TP}{TP + FN} \end{aligned}\right\}. \tag{1}$$

In general, high precision leads to low recall and vice versa. The $F_1$ score takes into account both precision and recall rate of the model and can be defined by Eq 2.

$$F_1 = \frac{2Pr \times Re}{Pr + Re}. \tag{2}$$

## Experimental results

**Experimental results on competition dataset.** As previously illustrated, there have been several approaches to solve the rebars recognition problem. Among of them, WSSKP [18] was a state-of-the-art algorithm, semantic segmentation and key point detection based on weak supervision were utilized in the model. The model achieved 98.8% $F_1$ score on DataFountain competition dataset. Fig 4 shows the comparison between WSSKP [18] and the proposed MaskID approach for two images. From Fig 4, it can be seen that one rebar is not recognized by WSSKP in each of the two images, as shown in Fig 4(a) and 4(c). For MaskID, all rebars were recognized, as shown in Fig 4(b) and 4(d). The model size, Precision, Recall and $F_1$-scores of the benchmark methods and MaskID for 200 testing images taken from the DataFountain competition dataset are shown in Table 2. It is noted that the results in Table 2 are averaged on 200 testing images. Because the results of benchmark methods are directly taken from the references, the Precision and Recall of these methods are not listed in Table 2 since these two metrics were not given in the original references. From this table, it can be seen that MaskID obtained the highest $F_1$-score (99.38%), which is a 0.58% improvement compared to the state-of-the-art WSSKP method [18].

**Experimental results on test platform dataset.** The performance of MaskID is also evaluated on our own experimental platform. In order to effectively detect multiscale rebars, the image is split into $384 \times 384$-pixel small images with 160 pixels overlapping. The recognition results obtained from MaskID is shown in Fig 5. The Precision, Recall and $F_1$ score are listed in Table 3. It can be seen that all three metrics are larger than 0.99. From Fig 5 and Table 3, it can be seen that MaskID has low miscounting errors, and even very small rebars with sizes below $8 \times 8$ pixels in the right-down region are accurately recognized.

## Summary

Accurate counting of densely-stacked rebars is essential in several real-world scenarios. Many deep learning-based methods, such as Faster R-CNN, Cascade R-CNN, and WSWA-Reg, have

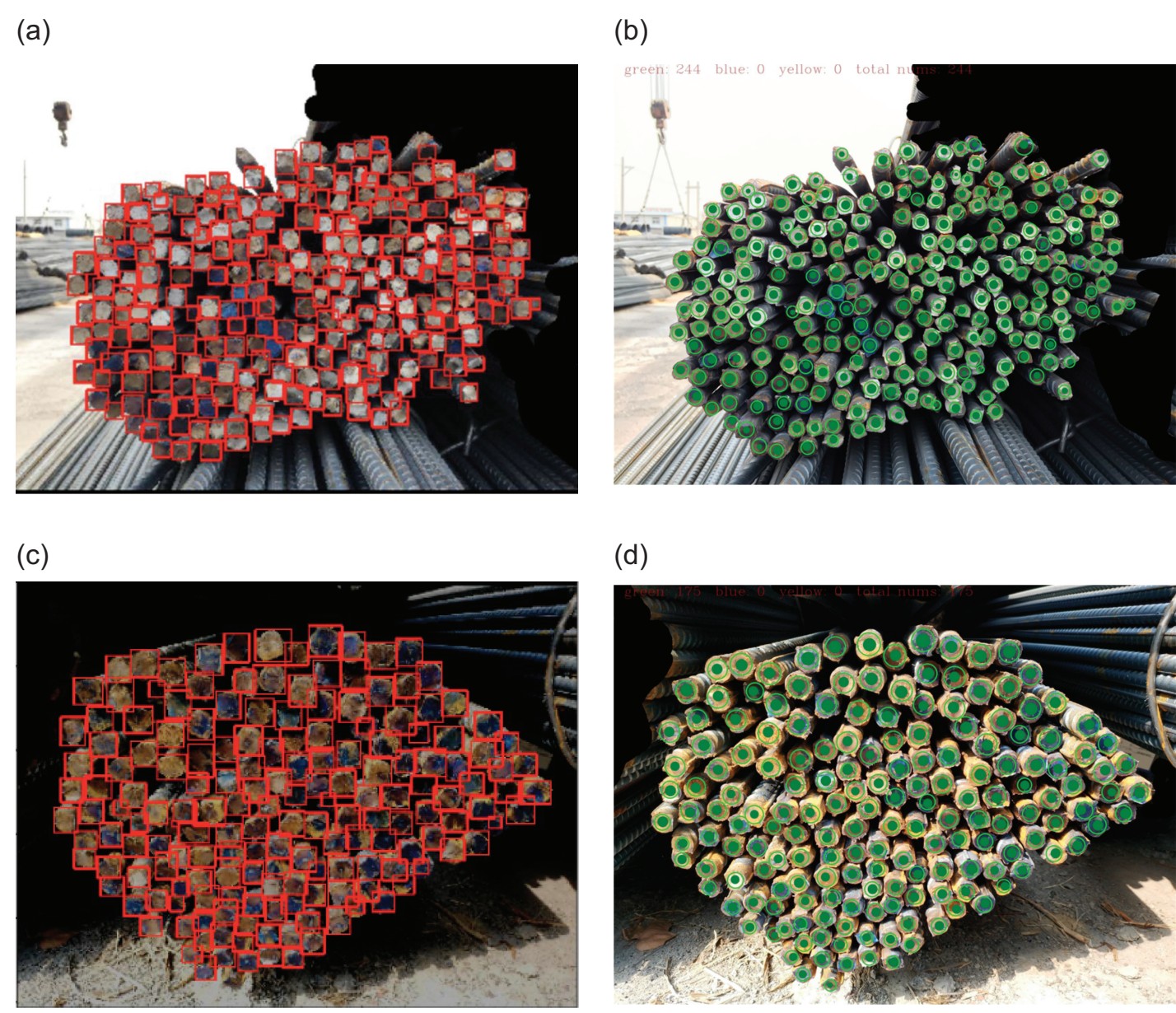

**Fig 4. (Color online)Recognition results on two images.** (a) and (c) show the results of WSSKP, and (b) and (d) show the results of MaskID.

**Table 2. Average Precision, Recall and $F_1$-score on 200 testing images from DataFountain competition dataset.**

|  | Model size | Precision | Recall | $F_1$-Score |
|---|---|---|---|---|
| Retina Net [9] | 16.3M | - | - | 0.9802 |
| Faster RCNN [4] | 23.2M | - | - | 0.9830 |
| Cascade RCNN [22] | 42.1M | - | - | 0.9870 |
| WSWA-Seg [23] | 5.8M | - | - | 0.9883 |
| WSSKP [18] | 9.2M | - | - | 0.9880 |
| MaskID | 10.1M | 0.9934 | 0.9942 | 0.9938 |

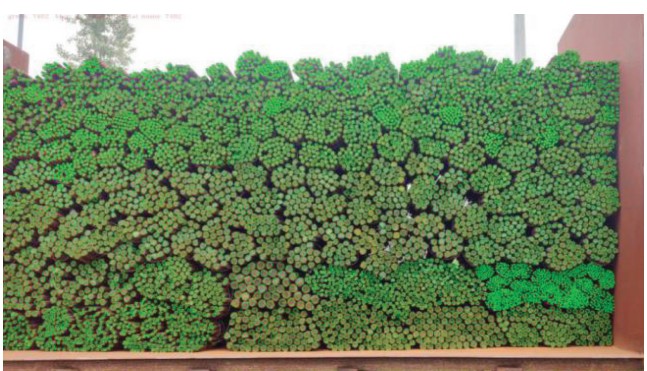

**Fig 5. (Color online) recognition results on platform dataset.**

**Table 3. The rebar counting results on test platform dataset.**

| The number of real rebars | Precision | Recall | $F_1$-Score |
|---|---|---|---|
| 7453 | 0.9978 | 0.9910 | 0.9944 |

been developed and have demonstrated effectiveness. However, there remain several challenges in counting densely-stacked rebars, as well as those with varied sizes and shapes. Based on the characteristics of the rebar section and the specific case of densely stacked rebars, a novel counting method based on deep learning and intelligent denoising strategies is proposed in this paper. First, improved Mask R-CNN was applied to obtain the segmentation results initially by utilizing a splitting and rescaling strategy. Then, intelligent denoising strategies corresponding to aspect ratio of recognized box, overlapping recognized objects, object distribution and environmental noise, were designed to eliminate background noise, including abnormal shape, reduplicative recognition, and distribution abnormality. The results show that the proposed method is capable of accomplishing the recognition and counting task with a score of over 0.9938 $F_1$-score, and outperforms state-of-the-art rebar recognition methods. The proposed method can be applied to building sites and rebar manufacturers for the rapid and accurate rebar counting. In this paper, fixed splitting and overlapping size are set in the proposed method. However, a rebar with certain real size might have different image size in different scene. So, the identifying results might unstable if the image size of rebar is particularly small or large. For example, the image size of rebar is larger than $400 \times 400$ pixels or smaller than $5 \times 5$ pixels, it is difficult to identify the rebars accurately utilizing the proposed algorithm with split window $384 \times 384$ pixels. In order to further improve the generalization ability of the algorithm, several splitting and overlapping size might be used to identify the rebars, and the best result will be adopted. Due to parallax, some rebars are completely covered by other rebars. For this specific case, a separate image can be retaken so that the covered rebars in the original image can be seen. Then stitch the recognition results of original image and retaken image to get the final results. In some ultra wide scenarios, it is necessary to take multiple images, and recognition results of multiple images will be stitched to final results. These problems will be further studied deeply in the future.

## Supporting information

**S1 File.**
(RAR)

## Acknowledgments

We thank Fei He and Haizhen Xie for data collection, and Wenyu Fu and Tianxiang He for their helpful discussion. We also thank reviewers for their constructive suggestions and comments, which have provided great help for us to improve the paper.

## Author Contributions

**Conceptualization:** Wenrui Li, Jian Cheng, Duanbing Chen.

**Data curation:** Wenrui Li, Bo Chen, Yu Xue, Yi Wang, Duanbing Chen.

**Formal analysis:** Wenrui Li, Duanbing Chen.

**Funding acquisition:** Duanbing Chen.

**Investigation:** Wenrui Li, Bo Chen, Yu Xue, Yi Wang, Yan Fu, Duanbing Chen.

**Methodology:** Wenrui Li, Yan Fu, Duanbing Chen.

**Project administration:** Jian Cheng.

**Validation:** Jian Cheng, Duanbing Chen.

**Visualization:** Bo Chen, Yi Wang, Duanbing Chen.

**Writing – original draft:** Wenrui Li, Duanbing Chen.

**Writing – review & editing:** Wenrui Li, Jian Cheng, Bo Chen, Yu Xue, Yi Wang, Yan Fu, Junlin Zhou, Duanbing Chen.

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
