## [Decision Letter · Decision Letter 0]

28 Mar 2022

PONE-D-21-28703MaskID: an effective deep-learning-based algorithm for dense rebar countingPLOS ONE

Dear Dr. Chen,

Thank you for submitting your manuscript to PLOS ONE. After careful consideration, we feel that it has merit but does not fully meet PLOS ONE’s publication criteria as it currently stands. Therefore, we invite you to submit a revised version of the manuscript that addresses the points raised during the review process.

We look forward to receiving your revised manuscript.

Kind regards,

Humaira Nisar

Academic Editor

PLOS ONE

Journal Requirements:

2. Thank you for stating in your Funding Statement: "This work was partially supported by the National Natural Science Foundation of China (Grant No. 61673085) by the Science Strength Promotion Program of UESTC under Grant No. Y03111023901014006."

5. We note that Figure 5 in your submission contain copyrighted images. All PLOS content is published under the Creative Commons Attribution License (CC BY 4.0), which means that the manuscript, images, and Supporting Information files will be freely available online, and any third party is permitted to access, download, copy, distribute, and use these materials in any way, even commercially, with proper attribution. For more information, see our copyright guidelines: http://journals.plos.org/plosone/s/licenses-and-copyright.

a. You may seek permission from the original copyright holder of Figure 5 to publish the content specifically under the CC BY 4.0 license. 

Reviewers' comments:

Reviewer's Responses to Questions

**Comments to the Author**

1. Is the manuscript technically sound, and do the data support the conclusions?

Reviewer #1: Yes

Reviewer #2: Yes

2. Has the statistical analysis been performed appropriately and rigorously? 

Reviewer #1: Yes

Reviewer #2: Yes

3. Have the authors made all data underlying the findings in their manuscript fully available?

Reviewer #1: Yes

Reviewer #2: Yes

4. Is the manuscript presented in an intelligible fashion and written in standard English?

Reviewer #1: Yes

Reviewer #2: Yes

5. Review Comments to the Author

Reviewer #1: The manuscript has presented a deep learning-based method for dense rebar counting. The manuscript is well eligible for publication because of the following reasons:

1. The case study and literature are well documented and drafted with a suitable English language.

2. The performance of the proposed method is accurate and outperformed the state-of-the-art methods.

Therefore, I would like to recommend the manuscript for acceptance in its present form.

Reviewer #2: This paper proposes MaskID, an innovative counting method based on deep learning and heuristic strategies. First, a sliced version of the Mask region-based convolutional neural network (R-CNN) was designed to obtain the segmentation results. Their extensive experimental results demonstrated that their approach reaches a high accuracy.

The proposed system is very interesting and quite straight-forward, however, there are still many important issues that should be addressed before final publication.

1. Authors may revise the abstract to elaborate more on problem statement and their findings and contributions.

2. Authors may elaborate more on the novelty of their work. How it contributes to the literature. In page 2, the authors write, "The two main principles of the approach are as follows." I don't see the novelty in the MaskID algorithm used. If no one has proposed before a method like the proposed algorithm, this claim should be highlighted much more. Else, it should be indicated who has done this, and it should be indicated what the innovations of the current paper are. Furthermore, briefly describe the major contributions in bullet form, just before the organization paragraph.

3. Introduction can be improved by having four clear and concise subsections on motivation of your research; background and related works; list of key contributions; and organization of the paper.

4. A comparative literature review may require identifying the problem/ research gap in Section "Materials and Methods". A few of the references are missing some information, may complete them critically. Provided references are better enough. However, authors are recommended to consider more latest and related such as,

https://www.sciencedirect.com/science/article/abs/pii/S1047320321001711

https://www.sciencedirect.com/science/article/abs/pii/S1389128619306978

https://ieeexplore.ieee.org/abstract/document/8606255.

5. What are the shortcomings of the algorithm and what is the room for improvement? I suggest adding a brief description in the conclusion section.

6. PLOS authors have the option to publish the peer review history of their article (what does this mean?). If published, this will include your full peer review and any attached files.

Reviewer #1: No

Reviewer #2: No

---

## [Author Response · Author response to Decision Letter 0]

13 May 2022

We have revised the manuscript according to editor's and reviewers' suggestions and comments. We have responded to the editor's and reviewers' suggestions or comments one by one in the response letter.

---

## [Decision Letter · Decision Letter 1]

30 May 2022

PONE-D-21-28703R1MaskID: an effective deep-learning-based algorithm for dense rebar countingPLOS ONE

Dear Dr. Chen,

Thank you for submitting your manuscript to PLOS ONE. After careful consideration, we feel that it has merit but does not fully meet PLOS ONE’s publication criteria as it currently stands. Therefore, we invite you to submit a revised version of the manuscript that addresses the points raised during the review process.

1) Section- Datasets: As mentioned by the authors 2 datasets have been used, it will be advisable if sample images from the 2 datasets should be shown to get an overview of the type of images. The first dataset has 200 images, but for the 2nd dataset, it is not clear, how many images are there. The results are only tabulated in the form of Table 2, using F1 score. The division of the datasets into training/validation and testing is not given. Nor the training and testing accuracy is mentioned. These parameters should be mentioned in the manuscript.

2) it looks like the combined results from the 2 datasets are given, which means that the 2 datasets were combined for the analysis, or dataset 2 was only used for testing and dataset 1 for training? Need to explain

3) In addition to the F-1 score it is advisable to include the values for precision and recall.

4) The details of the model should also be included.

5) Table 2 compares the result of the proposed algorithm with the state of the art. Are all these methods use the same datasets?

6) The references are a bit confusing, Ref 9 and Ref 21 are the same. It is mentioned as Retina Net, in which a dense detector is designed and named as retina NET. IT is not clear if the same dataset is used, then how to justify the results in Table 2.

7) Similar to above, ref 4 and Ref 22 are the same.

8) for table 2, if the same dataset is not used then the comparisons are not meaningful, it will be better if the authors give a detailed table in which they include the type of dataset the no of images, and results to get an overview of what is happening.

9)There are some spelling and grammatical mistakes that should be corrected. For example, line 33, should be scene.... line 85 novelty----novel.

We look forward to receiving your revised manuscript.

Kind regards,

Humaira Nisar

Academic Editor

PLOS ONE

Additional Editor Comments:

Thank you for revising the manuscript.

I have the following comments:

1) Section- Datasets: As mentioned by the authors 2 datasets have been used, it will be advisable if sample images from the 2 datasets should be shown to get an overview of the type of images. The first dataset has 200 images, but for the 2nd dataset, it is not clear, how many images are there. The results are only tabulated in the form of Table 2, using F1 score. The division of the datasets into training/validation and testing is not given. Nor the training and testing accuracy is mentioned. These parameters should be mentioned in the manuscript.

2) it looks like the combined results from the 2 datasets are given, which means that the 2 datasets were combined for the analysis, or dataset 2 was only used for testing and dataset 1 for training? Need to explain

3) In addition to the F-1 score it is advisable to include the values for precision and recall.

4) The details of the model should also be included.

5) Table 2 compares the result of the proposed algorithm with the state of the art. Are all these methods use the same datasets?

6) The references are a bit confusing, Ref 9 and Ref 21 are the same. It is mentioned as Retina Net, in which a dense detector is designed and named as retina NET. IT is not clear if the same dataset is used, then how to justify the results in Table 2.

7) Similar to above, ref 4 and Ref 22 are the same.

8) for table 2, if the same dataset is not used then the comparisons are not meaningful, it will be better if the authors give a detailed table in which they include the type of dataset the no of images, and results to get an overview of what is happening.

There are some spelling and grammatical mistakes that should be corrected. For example, line 33, should be scene.... line 85 novelty----novel.

Reviewers' comments:

Reviewer's Responses to Questions

**Comments to the Author**

1. If the authors have adequately addressed your comments raised in a previous round of review and you feel that this manuscript is now acceptable for publication, you may indicate that here to bypass the “Comments to the Author” section, enter your conflict of interest statement in the “Confidential to Editor” section, and submit your "Accept" recommendation.

Reviewer #1: All comments have been addressed

Reviewer #2: All comments have been addressed

2. Is the manuscript technically sound, and do the data support the conclusions?

Reviewer #1: Yes

Reviewer #2: Yes

3. Has the statistical analysis been performed appropriately and rigorously? 

Reviewer #1: Yes

Reviewer #2: Yes

4. Have the authors made all data underlying the findings in their manuscript fully available?

Reviewer #1: Yes

Reviewer #2: Yes

5. Is the manuscript presented in an intelligible fashion and written in standard English?

Reviewer #1: Yes

Reviewer #2: Yes

6. Review Comments to the Author

Reviewer #1: The updates in the manuscript are satisfactory and the case study is suitable for the publication. I would like to recommend it for acceptance.

Reviewer #2: The revised manuscript is satisfactory and can be beneficial to readers. Therefore, I would like to recommend the publication.

7. PLOS authors have the option to publish the peer review history of their article (what does this mean?). If published, this will include your full peer review and any attached files.

Reviewer #1: No

Reviewer #2: No

---

## [Author Response · Author response to Decision Letter 1]

17 Jun 2022

Thank you very much for processing our manuscript entitled “MaskID: an effective deep-learning-based algorithm for dense rebar counting”. We have considered all the points and revised the manuscript accordingly. Enclosed please find a detailed response. For the sake of convenience, modifications are marked in red in the revised manuscript. Moreover, a clean version of revised manuscript is also re-submited.

---

## [Editor Report · Decision Letter 2]

23 Jun 2022

MaskID: an effective deep-learning-based algorithm for dense rebar counting

PONE-D-21-28703R2

Dear Dr. Chen,

We’re pleased to inform you that your manuscript has been judged scientifically suitable for publication and will be formally accepted for publication once it meets all outstanding technical requirements.

Kind regards,

Humaira Nisar

Academic Editor

PLOS ONE
---

## [Editor Report · Acceptance letter]

27 Jun 2022

PONE-D-21-28703R2 

MaskID: an effective deep-learning-based algorithm for dense rebar counting 

Dear Dr. Chen:

I'm pleased to inform you that your manuscript has been deemed suitable for publication in PLOS ONE. Congratulations! Your manuscript is now with our production department. 

Kind regards, 

on behalf of

Dr. Humaira Nisar 

Academic Editor

PLOS ONE